# Acute Kidney Injury in a Cohort of Critical Illness Patients Exposed to Non-Steroidal Anti-Inflammatory Drugs

**DOI:** 10.3390/ph15111409

**Published:** 2022-11-14

**Authors:** Henry Oliveros, Giancarlo Buitrago

**Affiliations:** 1PhD Program in Clinical Epidemiology, Department of Clinical Epidemiology and Biostatistics, Pontificia Universidad Javeriana, Bogotá 110231, Colombia; 2School of Medicine, Universidad de La Sabana, Chía 250001, Colombia; 3Clinical Research Institute, School of Medicine, Universidad Nacional de Colombia, Bogotá 111321, Colombia

**Keywords:** nephrotoxicity, non-steroidal anti-inflammatory drugs, acute kidney injury (AKI), critical illness, negative control

## Abstract

To determine whether non-steroidal anti-inflammatory drug (NSAIDs) exposure prior to intensive care unit (ICU) admission affects the development of acute kidney injury (AKI) with renal replacement therapy (RRT). An administrative database is used to establish a cohort of patients who were admitted to the ICU. The exposure to NSAIDs that the patients had before admission to the ICU is determined. Demographic variables, comorbidities, AKI diagnoses requiring RRT, and pneumonia during the ICU stay are also measured. Multivariate logistic regression and inverse probability weighting (IPW) are used to calculate risks of exposure to NSAIDs for patients with AKI requiring RRT. In total, 96,235 patients were admitted to the ICU, of which 16,068 (16.7%) were exposed to NSAIDs. The incidence of AKI with RRT was 2.71% for being exposed to NSAIDs versus 2.24% for those not exposed (*p* < 0.001). For the outcome of AKI, the odds ratio weighted with IPW was 1.28 (95% CI: 1.15–1.43), and for the outcome of pneumonia as a negative control, the odds ratio was 1.07 (95% CI: 0.98–1.17). The impact of prior exposure to NSAIDs over critically ill patients in the development of AKI is calculated as 8 patients per 1000 exposures. The negative control with the same sources of bias did not show an association with NSAID exposure.

## 1. Introduction

Non-steroidal anti-inflammatory drugs (NSAIDs) are one of the most widely prescribed medications worldwide. Between 20% and 30% of patients over 65 years of age consume NSAIDs [1]. Each year in the United States, 50 million people spend $5–10 billion on prescription and over the counter (OTC) NSAIDs. Among the most frequent adverse effects are peptic ulcer disease, with a prevalence of 15–20% [2,3,4], while renal toxicity due to NSAID use has been reported less often. The two mechanisms involved are (i) the inhibition of prostaglandins, which decreases vasodilatation of the afferent arteriole in situations that need improved renal perfusion, and (ii) a direct inflammatory effect called interstitial nephritis [5,6,7,8]. Some studies have determined that mortality increases among patients with chronic kidney disease and NSAID consumption [9,10,11]. Nevertheless, all of these studies have been observational, and their demonstration of a causal association has been limited [12,13,14].

Patients admitted to intensive care units present different conditions that could lead to acute kidney injury, the most common being decreased renal function reserve, which is diminished by coexisting diseases, such as diabetes and hypertension [15,16]. The second most frequent condition is a disease that leads to intensive care, sepsis, trauma, heart failure, and shock states in general [8,17].

Few studies have explored the effects of NSAIDs in patients with more than one condition that impairs renal function, determining the effects of NSAID exposure should ideally be studied through a clinical trial, however, ethical limitations, costs and logistics prevent it. Having a large volume of administrative data has made it possible to use this information to study exposures in a large number of patients. The major limitation of observational studies is the systematic differences between the groups being compared. In recent years, causal inference methods have been used, such as the calculation of the propensity index and the weighting by the inverse weight of the propensity index, which allow to balance the differences in the basic characteristics between the comparison group. In this way, we can control the effect of confounding variables [18,19,20].

We wanted to assess the effect of NSAID exposure on the development of acute renal failure requiring renal replacement therapy in a cohort of patients admitted to intensive care units (ICUs) for any cause by using administrative data.

## 2. Results

Out of 99,407 patients whose first admission to the intensive care unit occurred between 1 January and 31 December 2018, a total of 3172 were excluded due to requiring renal replacement therapy during the previous two years (2016 and 2017), with 96,235 patients remaining (Figure 1). We found that 16,068 patients (16.7%) were exposed to NSAIDs during the 90 days prior to ICU admission, having taken at least one formulation each month. The incidence of acute renal failure (AKI) during ICU stay was 2.71% for the group of patients exposed to NSAIDs versus 2.24% for those not exposed (*p* < 0.001).

The coexisting diseases with the highest prevalence were hypertension (74.02%), diabetes (31.67%), and chronic pulmonary disease (27.53%). An imbalance between patients exposed to NSAIDs and those not exposed was observed in most of the baseline characteristics, as shown by the absolute standardized differences of over 10% (see Table 1). Baseline characteristics were balanced through propensity score matching, which resulted in a Rubin index of 2.7%, as shown in Table 2 and Figure 2 [21].

The odds ratios of an AKI outcome with renal replacement therapy (RRT) were calculated using multivariate logistic regression analysis, and the variables that were statistically significant for increased risk were, first, sepsis as a reason for admission to ICU OR of 2.99 (95% CI: 2.70–3.32), and second, the coexistence of liver disease OR of 2. 27 (95% CI: 1.70–3.03), followed by diabetes OR 1.99 (95% CI: 1.82–2.19), heart failure OR 1.90 (95% CI: 1.58–2.29), and hypertension OR 1.95 (95% CI: 1.69–2.25). Age over 60 years, female sex, and exposure to NSAIDs also showed an increased risk of AKI with RRT. Variables that were not associated with the presentation of acute kidney injury were presentation of pneumonia OR 1.28 (95% CI: 0.73–2.27), history of cerebrovascular disease OR 0.91 (95% CI: 0.77–1.07), connective tissue diseases OR 0.79 (95% CI: 0.66–0.95), ICU admission for trauma OR 0.77 (95% CI: 0.64–0.92), ICU admission for myocardial infarction OR 0.79 (95% CI: 0.70–0.89), history of peptic ulcer disease OR 0.71 (95% CI: 0.61–0.82), and chronic obstructive pulmonary disease OR 0.55 (95% CI: 0.50–0.62) (see Figure 3).

### 2.1. Association between NSAIDs, AKI, and Residual Confounding

In terms of the association between exposure to NSAIDs and AKI, the crude odds ratio was 1.21 (95% CI: 1.09–1.35), the odds ratio adjusted for sociodemographic characteristics and for coexisting diseases by multivariate logistic regression analysis was 1.25 (95% CI: 1.12–1.39), and the odds ratio obtained by inverse probability weighting was 1.28 (95% CI: 1.15–1.43). In terms of exposure to NSAIDs and the negative outcome of presenting with pneumonia, the crude odds ratio was 1.28 (95% CI: 1.18–1.38), the odds ratio adjusted by multivariate analysis was 1.10 (95% CI: 1.00–1.21), and with IPW, it was 1.07 (95% CI: 0.98–1.17) (see Table 3 and Appendix A).

### 2.2. Effect of NSAIDs on Presenting AKI

After matching based on the propensity score, the calculation of the difference between the average effect on those exposed to NSAIDs and those not exposed showed an incidence of AKI with RRT of 2.7% for the former and 1.9% for the latter, that is, a difference of 0.8%.

### 2.3. 28-Day Mortality of AKI and RRT Patients

The 28-day mortality was 11.1% overall, 14.25% for the group of patients exposed to NSAIDs, and 11.94% (*p* < 0.001) for those not exposed to NSAIDs, with a crude OR of 1.22 (95% CI: 1.16–1.28) and an adjusted OR of 1.41 (95% CI: 1.33–1.48), respectively.

The 28-day mortality for patients who presented kidney injury with RRT was 37.98% compared to 11.71% for patients who did not develop acute kidney injury (*p* < 0.001). The crude odds ratio was 4.6 (95% CI: 4.22–5.04), and the adjusted OR was 3.71 (95% CI: 3.37–4.08) (Appendix A).

## 3. Discussion

This study found an increased risk of acute kidney injury requiring renal replacement therapy when there was exposure to NSAIDs during a 90-day window prior to ICU admission. The crude odds ratio, adjusted odds ratio, and odds ratio obtained with IPW were consistent. The adjusted odds ratios included coexisting illnesses, demographic characteristics, and reasons for ICU admission.

Several studies have reported an increased risk of kidney disease with chronic NSAID use [22,23]. Some have also found an increased risk of mortality with exposure to NSAIDs for patients with established chronic renal failure. Ka Man Lai et al., determined 5-year mortality in a cohort of 3383 patients who were diagnosed with end-stage renal disease and found an adjusted OR 1.39 (95% CI: 1.21–1.60), with exposure to NSAIDs 90 days prior to requiring dialysis [13]. Given that there is evidence to support that AKI is a condition that increases all-cause mortality, AKI is an intermediate outcome in the causal pathway between exposure to NSAIDs and mortality (see Figure 2).

The study herein established that exposure to NSAIDs poses an indirect risk of mortality through the presentation of AKI requiring RRT and found 1.2% more AKI with RRT for patients exposed to NSAIDs. Although this percentage is low, mortality from kidney injury with RRT is very high, ranging from 34% to 50%. This is very similar to the 38% found by the present study, and the 32% to 56% range reported in the literature by Torgeir Folkestad et al., in a meta-analysis of 16 studies (1872 AKI patients) [24].

### Variables Associated with NSAID Use

As shown in Appendix A, the variables associated with an increased risk of exposure to NSAIDs were trauma, sepsis, stroke, and cancer, while the variables associated with a decreased risk of NSAID exposure were male sex and COPD. Since NSAID toxicity inhibits the production of prostaglandins, which produce vasodilatation of the afferent arteriole and thereby improve renal perfusion in situations of shock, it can be inferred that critical patients who are exposed to NSAIDs could have an increased risk of kidney injury when also presenting sepsis as a reason for ICU admission, previous liver disease, diabetes, heart failure, or hypertension, or when being male and of advanced age. However, connective tissue diseases, myocardial infarction, trauma, peptic ulcer disease, and chronic obstructive pulmonary disease did not increase the risk of acute kidney injury.

A limitation of the present study is the lack of information about severity scores in the population of patients admitted to the ICU, which would have provided information about the acuteness of the disease, thereby making it possible to adjust for severity at the time of admission to intensive care [25,26]. However, since this study included a large number of comorbidities in the Charlson index (validated version for Colombia) [27], and it took into account patient diagnoses when admitted to intensive care, as well as negative outcomes, such as pneumonia, we were better able to control for and diagnose residual confounding [28,29]. This resulted in less biased estimates, as shown by the comparison of crude and adjusted estimates of odds ratios obtained with multivariate analysis and IPW (see Table 3).

Another limitation was the determination of NSAID exposure in terms of the approach of using a formulation without evidence of adherence to treatment. However, the lack of records on exposure to non-prescribed and OTC NSAIDs would shift the results towards the null hypothesis by underestimating their consumption.

The strength of this study is the use of a large sample size and the inclusion of a large number of clinical and demographic characteristics associated with renal reserve in patients prior to ICU admission. This allowed us to obtain less biased estimates, as shown by the comparison of crude and adjusted estimates, as well as by the lack of association with the negative control.

Lastly, we wanted to evaluate the consistency between the different methods in relation to the calculation of the risk ratios. Then with the IPW methodology we obtained the most conservative estimators of the risk coefficients.

## 4. Materials and Methods

### 4.1. Type of Study and Sources of Information

A retrospective cohort study was conducted based on administrative data from the Integrated Social Protection Information System (SISPRO in Spanish) of the Colombian Ministry of Health. This database contains reports from the services that insurers (EPS in Spanish) provided to 22.4 million Colombians (45% of the population) who were enrolled in Colombia’s contributory health system in 2018, and which were paid under the health system’s benefits plan. The database is highly standardized and contains the codes for the services provided (CUPS in Spanish), the date on which the services were provided, age, sex, insurer, municipality, ICD-10 code, drugs prescribed, and the cost of the service. Information on mortality, date of death, and diagnoses associated with the cause of death was obtained from death certificates (RUAF in Spanish). The database has the identification of each anonymized patient and is joined by the key (ID).

### 4.2. Population

The study population consisted of Colombian patients over 18 years of age whose first admission to the ICU for any cause occurred between 1 January 2018, and 31 December 2018. Patients were identified by procedure codes for ICU hospitalization (see Appendix A). Excluded were those who required renal replacement therapy during the year prior to admission in any of the hemodialysis or peritoneal dialysis modalities.

### 4.3. Study Variables

The main exposure variable was a prescription for any type of NSAID (COX-1 or COX-2) for at least one formulation per month during the 90 days prior to ICU admission (see Appendix A). We began by evaluating the association up to a period of 12 months previous to admission to ICU, with which we did not find differences between the exposed and non-exposed to the NSAID. In relation to the outcome of kidney failure, we found a high prevalence of the exposition to the NSAID, and then we gradually decreased the window period according to the association and to biological plausibility. Patients who did not meet this NSAID exposure criterion were considered to belong to the unexposed cohort.

The main outcome was the presentation of kidney injury requiring renal replacement therapy during the ICU stay. This information was obtained from the UPC database based on the codes used to record dialysis procedures. Other outcomes that were included were mortality 28 days after ICU admission and the presentation of pneumonia during the ICU stay. The latter was used as a negative control since a causal association was not expected between exposure to NSAIDs and the development of pneumonia [28,29,30].

Other variables took into consideration demographic characteristics such as age, sex and place of origin. It also includes other characteristic such as the coexisting diseases in the Charlson index version for Colombia [21]. The information was also obtained from the UPC databases for the years 2016 and 2017, two years prior to admission to the ICU.

Other variables included were demographic characteristics, such as age, sex, and place of origin. Coexisting diseases in the Charlson index and validated for the Colombian population were also identified [27]. In addition, the study took into account information from the databases on coexisting diseases during the two years prior to ICU admission for the years 2016 and 2017. The causes of admission to the ICU include: (1) trauma, (2) heart failure, (3) sepsis, and (4) myocardial infarction was obtained based on algorithms constructed using CUPS procedure codes, ICD-10 diagnoses, and medication records from 2018. The first ICU admission was used in cases where the same patient was admitted several times (see Appendix A).

### 4.4. Statistical Analysis

First, baseline demographic and clinical characteristics of the NSAID-exposed group and the unexposed group were compared by calculating standardized differences to assess the balance between the two groups. Second, crude odds ratios were calculated between NSAID exposure and the outcomes of acute kidney injury and pneumonia. The latter was used to assess the degree of residual confounding [28].

To control for confounding from systematic differences between the groups to be compared, acute kidney injury (AKI) odds ratios were calculated using a multivariate logistic regression analysis and adjusting for demographic characteristics, coexisting diseases, and conditions at the time of ICU admission.

To evaluate the consistency of the multivariate adjustment of the risk coefficients, additional weighting techniques were performed by calculating the propensity index (PPI), which was obtained from a logistic regression model based on the following variables: age, sex, hypertension, diabetes, cerebrovascular disease, connective tissue disease, liver disease, cancer, presence of arterial hypertension, and condition upon admission to the ICU, which included trauma, sepsis, myocardial infarction, and congestive heart failure. Standardized differences of less than 0.1 and the lowest Rubin index value were the criteria used for the best balance of baseline characteristics between the groups [21]. Standard errors were calculated by estimating robust standard errors. AKI and pneumonia odds ratios with NSAID exposure were recalculated with inverse probability weighting according to recommendations by Austin [18,19,31].

## 5. Conclusions

The acute impact of exposure to NSAIDs in critically ill patients prior to admission to the ICU has been established by finding an increased risk in the odds ratios that were calculated. Furthermore, we found that 0.8% more patients who used NSAIDs presented with AKI comorbid with RRT than patients who did not use NSAIDs. The use of a negative control, such as the outcome of pneumonia, showed no association with NSAID exposure or adequate control of residual confounding.

## Figures and Tables

**Figure 1 pharmaceuticals-15-01409-f001:**
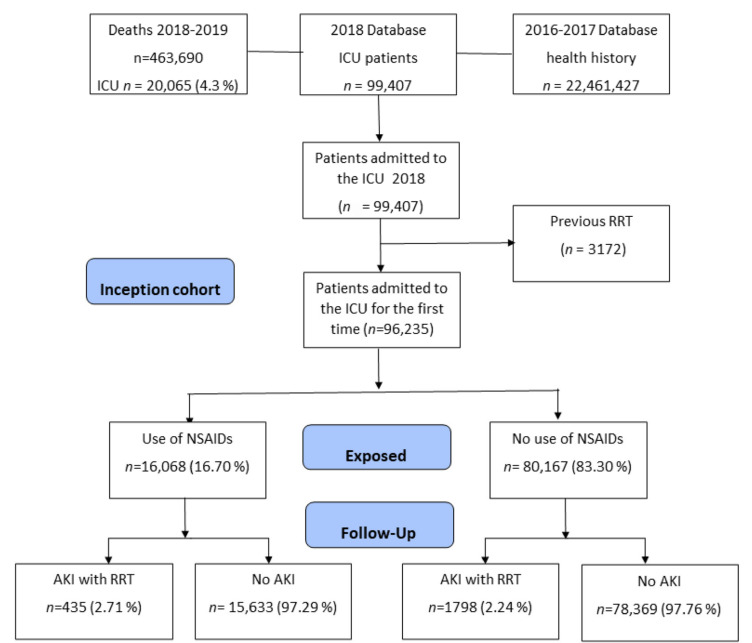
Flowchart of subjects included based on administrative health data.

**Figure 2 pharmaceuticals-15-01409-f002:**
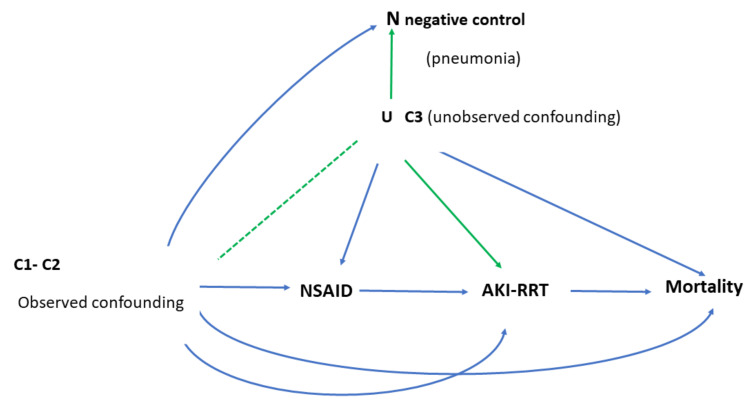
Causal pathway between NSAIDs and AKI and the relationship with confounding variables. **C1**: trauma, sepsis, myocardial infarct, congestive heart failure; **C2**: age, sex, hypertension, diabetes cerebrovascular disease, chronic pulmonary disease, peptic ulcer disease, liver disease; **C3:** liquid balance, hypotension, vasopressors.

**Figure 3 pharmaceuticals-15-01409-f003:**
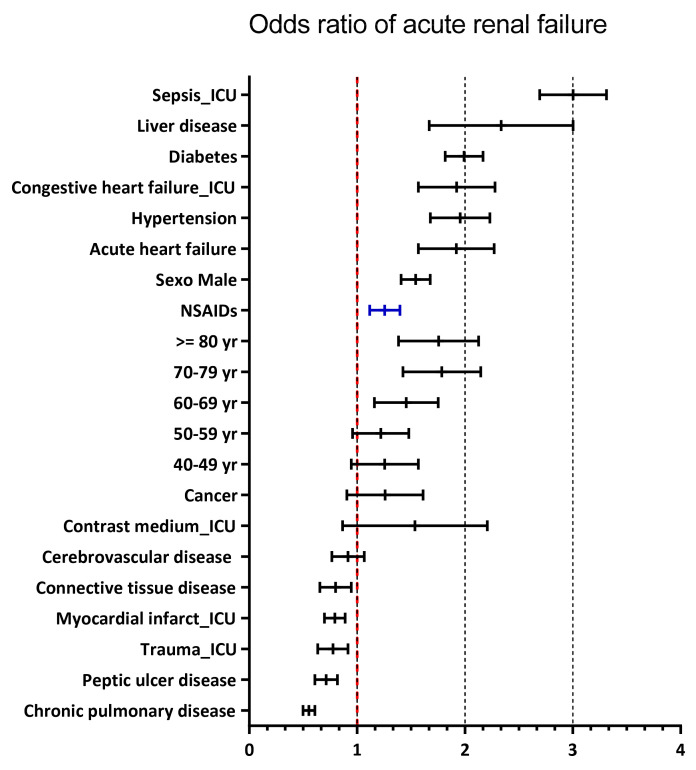
Factors for the development of acute kidney injury.

**Table 1 pharmaceuticals-15-01409-t001:** General characteristics of patients admitted to the ICU from 2018 in 180 units in Colombia.

		Exposure Status		
Characteristic No. (%)	Full Sample	NSAID	Not NSAID	Standardized Differences %
	*n* = 96,235 (100)	*n* = 16,068 (16.70)	*n* = 80,167 (83.30)	
Median age (IQR)—year	65 (53–76)	61 (47–73)	66 (54–76)	−25
Age group—no. (%)				
18–39 yr.	11,770 (12.23)	2924 (18.20)	8846 (11.03)	20
40–49 yr.	7702 (8.0)	1546 (9.62)	6156 (7.68)	6.9
50–59 yr.	16,011 (16.64)	2850 (17.74)	13,161 (16.42)	3.5
60–69 yr.	23,193 (24.10)	3597 (22.39)	19,596 (24.44)	−4.9
70–79 yr.	21,631 (22.48)	3052 (18.99)	18,579 (17.25)	−10.3
≥80 yr.	15,928 (16.55)	2099 (13.06)	13.829 (17.25)	−11.7
Sex—no. (%)				
Male	46,604 (48.43)	6905 (42.97)	39,699 (49.52)	−13.2
Female	46,631 (51.57)	9163 (57.03)	40,468 (50.48)	
ICU admission				
Trauma	6536 (6.79)	1382 (8.60)	5154 (6.43)	8.2
Sepsis	8739 (9.08)	1894 (11.79)	6845 (8.54)	10.8
Myocardial infarction	13,778 (14.32)	2037 (12.68)	11,741 (14.65)	−5.7
Congestive heart failure	2664 (2.77)	392 (2.44)	2272 (2.83)	−2.5
Contrast medium	590 (0.61)	140 (0.87)	450 (0.56)	3.7
Coexisting conditions—no. (%)				
Hypertension	71,232 (74.02)	11,347 (70.62)	59,885 (74.70)	−9.2
Diabetes	30,478 (31.67)	4.699 (29.24)	25,779 (32.16)	−6.3
Cerebrovascular disease	6314 (6.56)	1181 (7.35)	5133 (6.40)	3.7
Chronic pulmonary disease	26,490 (27.53)	3581 (22.29)	22,909 (28.58)	−14.5
Connective tissue disease	7056 (7.33)	1211 (7.54)	5845 (7.29)	0.9
Peptic ulcer disease	10,901 (11.33)	1823 (11.35)	9078 (11.32)	0.1
Liver disease	968 (1.01)	155 (0.96)	813 (1.01)	−0.5
Cancer	2011 (2.09)	475 (2.96)	1536 (1.92)	6.8
Outcomes in ICU				
Acute kidney injury	21,233 (2.32)	435 (2.71)	1798 (2.24)	3.0
Pneumonia	3.836 (3.99)	775 (4.82)	3.061 (3.82)	4.9
28-day mortality	11,858 (12.32)	2289 (14.25)	9569 (11.94)	

**Table 2 pharmaceuticals-15-01409-t002:** Baseline characteristics for each of the groups, after matching.

Variable	Exposed to NSAIDs *n* = 16,068	Not Exposed to NSAIDs *n* = 80,167	Standardized Differences	*p* Value
Age group—no. (%)				
18–39 yr.	18.2	18.2	0.1	0.965
40–49 yr.	9.6	9.7	−0.4	0.706
50–59 yr.	17.7	17.6	0.2	0.884
60–69 yr.	22.3	22.4	−0.1	0.904
70–79 yr.	18.9	18.9	0.1	0.932
≥80 yr.	13.1	12.9	0.2	0.868
Sex—no. (%)				
Male	43	43	0.8	0.955
ICU admission				
Trauma	8.6	8.5	0.3	0.826
Sepsis	11.7	11.8	−0.1	0.904
Myocardial infarction	12.6	12.4	0.8	0.479
Congestive heart failure	2.4	2.3	0.4	0.688
Contrast medium	0.8	0.8	0.8	0.501
Coexisting conditions—no. (%)				
Hypertension	70.6	70.7	−0.3	0.825
Diabetes	29.2	29.1	0.2	0.825
Cerebrovascular disease	7.4	7.3	0.1	0.949
Chronic pulmonary disease	22.2	22.1	0.4	0.717
Connective tissue disease	7.5	7.4	0.4	0.751
Peptic ulcer disease	11.34	11.2	0.4	0.751
Liver disease	0.9	0.7	1.9	0.061
Cancer	2.9	2.7	1.1	0.385

Rubin index of 2.7%.

**Table 3 pharmaceuticals-15-01409-t003:** Risks of outcome with NSAID exposure.

Outcomes	OR (95% CI) Unadjusted	OR (95% CI) Adjusted *	OR (95% CI) IPW **
AKI with RTT	1.21 (1.09–1.34)	1.25 (1.12–1.39)	1.28 (1.15–1.43)
Pneumonia	1.28 (1.18–1.38)	1.10 (1.00–1.21)	1.07 (0.98–1.17)

*Adjusted by age, sex, trauma, sepsis, myocardial infarct, congestive heart failure, contrast medium, hypertension, diabetes, cerebrovascular disease, chronic pulmonary disease, peptic ulcer disease, liver disease, cancer. ** IPW: Inverse probability weighting.

## Data Availability

The Information System (SISPRO) of the Colombian Ministry of Health provided the anonymized databases. The refined bases and the code used in the creation of the algorithms were made in STATA MP V.16 and is available to anyone who requests it.

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
