# Peer review of "Acute Kidney Injury in a Cohort of Critical Illness Patients Exposed to Non-Steroidal Anti-Inflammatory Drugs"

_pharmaceuticals, 2022, doi:10.3390/ph15111409_

Round 1
Reviewer 1 Report
This is an interesting and well conducted study that I believe will be of interest of the readers of Pharmaceuticals. My only comments are written below.
Main comments:
1) Please, expand the introduction to justify the use of IPW and propensity scores and negative controls. Even though the readers may understand that confounding variables need to be controlled for, it is necessary to explain why these other causal methods are needed and/or are superior to other more traditional analytical strategies.
2) On the same lines, it is necessary to clarify why BOTH, IPW and propensity scores are used. Throughout the article it is sometimes confusing when the authors are using one or the other. Furthermore, in the abstract nothing is mentioned about propensity scores.
3) It would be good to justify better the definition of exposure from a biological point of view. For example: is one month of exposure enough? Would be better to consider 2 months, or would 15 days be enough? Maybe, it would be good doing some sensitivity analyses regarding the exposure definition if considered necessary.
4) There is not much difference between the traditional adjustment and the IPW use, so I miss some discussion on why it is so. This is also related to my first point about the need of better justifying the use of causal methods.
Minor comments:
- “an increased risk in the odds ratios” – an increased odds?
- In tables 1 and 2: What is the meaning of ----no?
- Table 3: there is one 0 extra and unnecessary for pneumonia in the adjusted model
Author Response
Response to Reviewer 1 Comments
Point 1: Please, expand the introduction to justify the use of IPW and propensity scores and negative controls. Even though the readers may understand that confounding variables need to be controlled for, it is necessary to explain why these other causal methods are needed and/or are superior to other more traditional analytical strategies.
Response 1: We have introduced this clarification in the introduction section and inserted: determining the effects of NSAID exposure should ideally be studied through a clinical trial, however, ethical limitations, costs and logistics prevent it. Having a large volume of administrative data has made it possible to use this information to study exposures in a large number of patients. The major limitation of observational studies is the systematic differences between the groups being compared. In recent years, causal inference methods have been used, such as the calculation of the propensity index and the weighting by the inverse weight of the propensity index, which allow to balance the differences in the basic characteristics between the comparison group. In this way, we can control the effect of confounding variables.
Point 2: On the same lines, it is necessary to clarify why BOTH, IPW and propensity scores are used. Throughout the article it is sometimes confusing when the authors are using one or the other. Furthermore, in the abstract nothing is mentioned about propensity scores.
Response 2: Thank you for the comment.
In order to clarify the methods used , it was clarified that the calculation of the propensity index was used only for the weighting through the IPW We have introduced this clarification in the statistical analysis section and have inserted:
To evaluate the consistency of the multivariate adjustment of the risk coefficients, additional weighting techniques were performed by calculating the propensity index (PPI), which was obtained from a logistic regression model based on the following variables:
Point 3: It would be good to justify better the definition of exposure from a biological point of view. For example: is one month of exposure enough? Would it be better to consider 2 months?, or would 15 days be enough? Maybe, it would be good doing some sensitivity analyses regarding the exposure definition if considered necessary.
Response 3:
We have introduced this clarification in the methods section and have inserted:
We began by evaluating the association up to a period of 12 months previous to admission to ICU, with which we did not find differences between the exposed and non-exposed to the NSAID. In relation to the outcome of kidney failure, we found a high prevalence of the exposition to the NSAID, and then we gradually decreased the window period according to the association and to biological plausibility.
Point 4: There is not much difference between the traditional adjustment and the IPW use, so I miss some discussion on why it is so. This is also related to my first point about the need of better justifying the use of causal methods.
Response 4 : We have introduced this clarification in the discussion section and have inserted:
We wanted to evaluate the consistency between the different methods in relation to the calculation of the risk ratios. Then with the IPW methodology we obtained the most conservative estimators of the risk coefficients.
Minor comments:
Point 1 “an increased risk in the odds ratios” – an increased odds?
Response 1 :
Thank you for the correction ,
In the manuscript we leave only an increased odds?.
Point 2
- In tables 1 and 2: What is the meaning of ----no?
Response 2 :
Changed "no" to " exposed to NSAIDs
not exposed to NSAIDs" in tables 1 and 2
Point 3
Table 3: there is one 0 extra and unnecessary for pneumonia in the adjusted model
Response 3:
Correction 1.10 (1.00-1.21) was made

Reviewer 2 Report
Dear Authors,
this is a very interesting paper regarding the effects of nonsteroidal anti-inflammatory drug (NSAIDs) that could affects the development of acute kidney injury 15 (AKI) with renal replacement therapy (RRT).
I have minor comments to address:
1) the title is too long, please reduce it in 15 words maximum
2) the study is a cohort study. It must be clearly stated into the title. In addition the study design and participants selection is not well described.
3) in table 1 instead of standardized difference, please report the CI 95%
and the p value
4) the discussion must emphatize more on the clinical characteristics of the patients. It is missing this part. For example, gender, age or BMI
5) also nutritional status could affect the kidney function, please discuss and cite this paper Perna, Simone, Fatima Faisal, Daniele Spadaccini, Tariq A. Alalwan, Zahra Ilyas, Clara Gasparri, and Mariangela Rondanelli. "Nutritional Intervention Effectiveness on Slowing Time to Dialysis in Elderly Patients with Chronic Kidney Disease—A Retrospective Cohort Study." Geriatrics 7, no. 4 (2022): 83.
Author Response
Response to Reviewer 2 Comments
Point 1: the title is too long, please reduce it in 15 words maximum
Response 1 Thanks for the suggestion.
We have changed the title as follows:
Acute kidney failure in a cohort of critical ill patients exposed to non-steroid anti-inflammatory drugs.
Point 2: the study is a cohort study. It must be clearly stated into the title. In addition the study design and participants selection is not well described.
Response 2: Thank you for your suggestion.
We have modified the title including the information of the cohort
Additionally, in the section on the population in the methods, the criteria for inclusion of the cohort are specified.
We wanted to assess the effect of NSAID exposure on the development of acute renal failure requiring renal replacement therapy in a cohort of patients admitted to intensive care units (ICUs) for any cause, using administrative data
Point 3: n table 1 instead of standardized difference, please report the CI 95%
and the p value
Response 3:
In table 1 we wanted to show the magnitude of the differences through the standardized differences without the effect of sample size, table 2 does show the p values once the groups have been balanced.
Point 4: the discussion must emphatize more on the clinical characteristics of the patients. It is missing this part. For example, gender, age or BMI
Response 4 : Unfortunately, we did not have access to the BMI information, since it is not usually recorded in the administrative databases, however, when adjusting for the main coexisting diseases contained in the Charlson co-morbidity index, an adequate balance was obtained in the baseline characteristics of BMI. each of the groups, which was observed in table 2, finally using a negative control as an outcome, in association with exposure to NSAIDs, allowed evaluating residual confusion
Point 5: also nutritional status could affect the kidney function, please discuss and cite this paper Perna, Simone, Fatima Faisal, Daniele Spadaccini, Tariq A. Alalwan, Zahra Ilyas, Clara Gasparri, and Mariangela Rondanelli. "Nutritional Intervention Effectiveness on Slowing Time to Dialysis in Elderly Patients with Chronic Kidney Disease—A Retrospective Cohort Study." Geriatrics 7, no. 4 (2022): 83.
Response 5
The study "Nutritional Intervention Effectiveness on Slowing Time to Dialysis in Elderly Patients with Chronic Kidney Disease" is very interesting. In this study it is determined how the nutritional status impacts the progression of renal function deterioration.
Our study includes a population of patients different from those included in the study by Simone Perna et al. We studied critically ill patients who were admitted to the intensive care unit and who developed acute renal failure requiring dialysis therapy. Although it is true that nutritional status may be associated with the progression of renal failure, our sources of information do not allow for us to extract the information on the nutritional status. However, we took into account the information on the coexistence of diseases that determine the previous renal reserve, and we also included the information on the reason for admission to the intensive care unit.
